# CRISPR-Cas System: The Current and Emerging Translational Landscape

**DOI:** 10.3390/cells12081103

**Published:** 2023-04-07

**Authors:** Narendranath Bhokisham, Ethan Laudermilch, Lindsay L. Traeger, Tonya D. Bonilla, Mercedes Ruiz-Estevez, Jordan R. Becker

**Affiliations:** Corporate Research Material Labs, 3M Center, 3M Company, Maplewood, MN 55144, USA

**Keywords:** CRISPR-Cas, translational research, clinical study, gene editing, base editors, prime editors, CRISPR animal models

## Abstract

CRISPR-Cas technology has rapidly changed life science research and human medicine. The ability to add, remove, or edit human DNA sequences has transformative potential for treating congenital and acquired human diseases. The timely maturation of the cell and gene therapy ecosystem and its seamless integration with CRISPR-Cas technologies has enabled the development of therapies that could potentially cure not only monogenic diseases such as sickle cell anemia and muscular dystrophy, but also complex heterogenous diseases such as cancer and diabetes. Here, we review the current landscape of clinical trials involving the use of various CRISPR-Cas systems as therapeutics for human diseases, discuss challenges, and explore new CRISPR-Cas-based tools such as base editing, prime editing, CRISPR-based transcriptional regulation, CRISPR-based epigenome editing, and RNA editing, each promising new functionality and broadening therapeutic potential. Finally, we discuss how the CRISPR-Cas system is being used to understand the biology of human diseases through the generation of large animal disease models used for preclinical testing of emerging therapeutics.

## 1. Introduction

This review will focus on the transformation of human medicine brought about by the CRISPR-Cas system. Since its inception in 2012, the Cas9 variant of the CRISPR-Cas system has generated enormous interest due to the ease with which any DNA sequence in the genome can be targeted for alteration. The CRISPR-Cas9 system comprises two elements: an RNA-guided DNA endonuclease called Cas9 isolated from *Streptococcus pyogenes* and its guiding RNA called guide RNA or gRNA [1]. Cas9 in combination with gRNA can be engineered to target genomic sequences that are complementary to the sgRNA and catalyze a double-stranded break (DSB) in the DNA backbone. DSBs are then primarily repaired by either the error-prone non-homologous end-joining (NHEJ) pathway [2], or by error-free homology-directed repair (HDR) [3,4]. These mechanisms can be exploited to either disrupt (knock-out) genes or introduce new sequences (knock-ins) into the host genome [5]. The versatility and ease of use of the CRISPR-Cas system represents a transformative improvement upon previous gene editing technologies (e.g., Transcription Activator-Like Effector Nucleases, TALEN) [6]. The ability to easily target any region of the genome and create knock-ins or knock-outs is valuable in multiple fields including human medicine [7], agriculture [8], industrial biotechnology [9], among others.

While CRISPR-Cas systems are revolutionizing medicine, these systems are themselves undergoing an evolution. The single greatest liability of the Cas9-based gene editing is the formation of off-target double-stranded breaks in the genome which have the potential to generate mutations, large chromosomal aberrations such as translocations, inversions, etc. [10]. To overcome this limitation, Cas9-based base editors and prime editors [11] have been developed that operate without inducing a double-stranded break, thereby reducing the risk of chromosomal rearrangements. In this article, we will cover the current state of the art pertaining to base and prime editing in the field of human medicine. In addition to base and prime editors, Cas9 has been mutated to create an enzymatically deactivated Cas9 or dead Cas9 (dCas9) that has no endonuclease activity yet retains the ability to specifically locate and bind onto a target DNA sequence [12]. Standalone dCas9s as well as dCas9 fusions with various transcriptional activators [13] have enabled the modulation of transcriptional activity. Though dCas9 is not part of any therapeutic product that is currently under development, dCas9 serves as a tool of academic interest enabling the screening of signaling pathways and helping further the understanding of human disease biology [14]. Further, newer Cas proteins are being evaluated that provide differentiated editing performance over the *S. pyogenes* Cas9, such as the ones that selectively allow the editing of RNA instead of DNA [15,16]. In our article, we collectively refer to all Cas variants as the CRISPR-Cas system, and we cover some of the relevant topics surrounding the use of Cas9, dCas9s and other novel Cas proteins in the context of translational medicine (Figure 1). Finally, another important aspect of translational medicine is the availability of disease models in animals to test the efficacy of various therapies prior to clinical trials in humans. CRISPR-Cas systems have enabled the creation of monogenic disease models in animals that can then act as test beds to study various therapies, including CRISPR-Cas-derived cell and gene therapies [17,18,19]. In this article, we will cover some of the latest advances in the creation of animal models using the CRISPR-Cas system-based gene editing tools.

Within its first decade [1], the Cas9-based CRISPR-Cas system has altered the limits of human medicine, with several ongoing clinical trials leveraging this technology. In the coming sections, we will discuss how various CRISPR-Cas systems are not only used to treat monogenic disorders such as sickle cell anemia and Duchenne muscular dystrophy but also complex heterogenous diseases such as cancer, HIV-AIDS, and diabetes. CRISPR-Cas systems have seamlessly integrated into the gene and cell therapy ecosystem that has been under development for over the three decades, enabling ex vivo gene editing in the form of cell therapy and in vivo gene editing in the form of gene therapy [20,21]. In this article, we will detail the status of several completed and ongoing clinical trials being carried out with various CRISPR-Cas systems for treating various human diseases.

## 2. Current Use of CRISPR-Cas Systems in Clinical Trials

As of 15 December 2022, we identified 71 different clinical trials in the clinicaltrials.gov database that were directly associated with various CRISPR-Cas systems. Of the 71 trials, we identified 45 ongoing or completed trials that are directed towards human therapeutic product development. Of these, 23 trials were on various forms of cancer; 14 trials were on blood-related disorders including sickle cell disease and β-thalassemia; 3 were trials on HIV infection; and 1 trial each was on HPV, viral keratitis (HSV-1), Leber congenital amaurosis, hereditary angioedema, and Duchenne muscular dystrophy. In the coming sections, we will discuss how CRISPR-Cas systems are being leveraged for therapeutic development for each of these conditions.

### 2.1. CRISPR-Cas Technologies in the Treatment of Cancers

All clinical trials registered so far for cancer treatment use the Cas9 variant of the CRISPR-Cas system and involve the knock-out (KO) of one or more selected genes of interest. A small number of trials use the CRISPR-Cas9 system to knock-in specified sequences at target loci, and to our knowledge, there is one trial (NCT05397184 [22]) employing the base editing technology. In the upcoming section, we will survey the current range of clinical trials that involve knock-in/out of genes using CRISPR-Cas9. The lone clinical trial related to base editing for cancer is covered in the later sections of the paper.

#### 2.1.1. Cellular Immunotherapies

CRISPR-Cas9 technology is being increasingly used for engineering T lymphocytes, a type of immune cell that plays a central role in the fight against cancer. In the last decade, a new class of therapies referred to as adoptive cell therapy or cellular immunotherapy has gained prominence, and this involves harvesting and reprogramming a patient’s own T cells and employing them to treat cancer [23,24]. One of the foremost approaches in the cellular immunotherapy field is the CAR-T therapy, where T cells from cancer patients are isolated, engineered with chimeric antigen receptors (CAR), and injected back into patients [25] (Figure 2). CARs are designed to specifically target select antigens that are presented on the cancer cells. As of December 2022, six CAR-T therapies have been approved by the FDA to target CD19 and BCMA antigens in B cell-related cancers [26]. In addition to CAR-T, other adoptive cell therapy approaches include engineering a synthetic T cell receptor (TCR) into T cells and using them for treating cancer, referred to as TCR Therapy or TCR-T [27]. Tumor-infiltrating lymphocytes (TILs) therapy is another approach where lymphocytes that have infiltrated the tumors are isolated, enriched with or without genetic modifications and later injected back into cancer patients [28]. Although most current clinical trials involving the CRISPR-Cas9 technology are focused towards CAR-T therapy, we will also highlight a few TCR-T- and TIL-based therapies that involve CRISPR-Cas9 engineering in the sections below (See Table 1 and Table 2).

CAR-T therapies have proven to be very effective against B cell malignancies by enabling complete remissions in >80% of patient populations; however, the median event-free survival rate after 12 months was only in the range of 40–60% [29,30], indicating the occurrence of cancer relapse in patients who had complete remission earlier. Some of the drawbacks of the current class of CAR-T therapies include antigen escape by the cancer cells where CAR-specific antigens are lost in the cancer cells due to the selective pressure posed by the CAR-T, eventually leading to the relapse of cancer cells without the targeted antigen [31]. Beyond B cell malignancies, CAR-T treatments have proven to be less effective against solid cancers due to a variety of reasons including the presence of strong immunosuppressive microenvironments in solid cancers, inability of the engineered T cells to infiltrate into them and a lack of uniform presentation of cancer-specific target antigens across all cells in a solid tumor [32,33]. Due to the onset of exhaustion or dysfunction in T cells that is mediated by internal signaling mechanisms, CAR-T cells have also been unable to display anti-tumor activity for longer periods of time [34]. In the case of T cell malignancies, CAR-T have proven to be more ineffective due to the difficulty in separating a healthy T cell from a tumor-infected T cell. In addition, due to the presence of CAR-specific antigens in both healthy T cells and CAR-T cells themselves, T cell aplasia and fratricide are some of the risks associated with employing the T cell therapy towards T cell malignancies [35]. A variety of approaches are being pursued to alleviate many of the drawbacks listed above and are detailed in many reviews [36,37]. In this article, we will focus specifically on clinical trials where the CRISPR-Cas9 technology has been employed to alleviate some of the drawbacks of the current generation of T cell therapies.

#### 2.1.2. Role of CRISPR-Cas9 Tools to Mediate Immune Check Point Inhibition in CAR-T Therapy

Persistent antigen stimulation in T cells coupled with immunosuppressive microenvironments in cancers lead to a loss of anti-tumor activity in T cells, referred as T cell exhaustion or dysfunction [34]. One strategy to increase the anti-tumor activities of T cells and delay the exhaustion of T cell function is to use anti-programmed cell death protein (PD1) therapy [38]. PD1 protein (coded by *PDCD1*) expressed on the surface of the T cells and is a key immune check point regulator. Upon PD1′s interaction with its ligand, the programmed death ligand (PDL1) that is expressed on surfaces of cancer cells, PD1 mediates a negative regulatory role in the anti-tumor activity of T cells [39]. Blocking PD1′s interaction with PDL1 ligand via the anti-PD1 monoclonal antibody therapy has proven to be effective in combination with various T cell therapies including CAR-T therapy [40].

With the advent of facile gene editing technologies, CRISPR-Cas9 has been used for disruption of the *PDCD1* gene encoding for PD1, and this has resulted in the enhancement of anti-tumor effects of CAR-T in animal models [41,42]. Consequently, several clinical trials [43,44,45,46,47,48,49,50] have been registered that involve using the CRISPR-Cas9 system for disruption of *PDCD1* in CAR-T and TCR-T cells. So far, the results from three phase 1 trials have been published [51,52,53], but the results are inconclusive for the enhanced anti-tumor effect produced by the disruption of *PDCD1*. In one trial [51], T cells carried disruptions in *PDCD1* and endogenous TCR loci (*TRAC* and *TRBC*). In addition, these cells were also inserted with exogenous TCRs specific to NY-ESO-1 tumor antigen. These CRISPR-Cas9-engineered T cells were found to be persisting at peak levels for 4 months after infusion and displayed anti-tumor activity even for up to 9 months post infusion in one of the patients in the trial. Persistence of engineered T cells in this study was longer than in previous trials where similar cells without disruptions to *PDCD1* and endogenous TCR loci peaked after a week of injection and had a half-life of ~1 week [54,55,56]. The reasons for prolonged persistence in this study could be attributed to disruption in *PDCD1* and/or endogenous TCR genes, or other external factors such as exogenous T cell receptor design, etc. The exact reasons for enhanced survival of engineered T cells needs to be studied further.

In another trial [53], mesothelin-directed CAR-T cells were used to target mesothelin-positive tumors, and here, these cells had disruptions in the *PDCD1* locus as well as in the *TRAC* locus of TCR. Contrary to the previous study discussed above [51], in this study, PD1-disrupted CAR-T cells did not display prolonged persistence and enhanced anti-tumor activity in patients. Levels of engineered T cells peaked a week after injection and became undetectable after 1 month. In a similar study involving the mesothelin-directed CAR-T cells with intact *PDCD1*, when used in conjunction with the anti-PD1 therapy, they displayed prolonged persistence [57]. The reasons for the lack of prolonged persistence in this study could be multifactorial, but an interesting observation that the authors point to was that a minority of injected cell populations that had their endogenous TCR intact were found to be persisting at the end of the study, hinting at the possibility of the importance of TCR signaling playing a role in prolonged persistence of these engineered cells. In the third study [52], while PD1-disrupted cells persisted in patients for 4 four weeks after infusion, no objective conclusions could be made on the effect of *PDCD1* disruption on anti-tumor activity due to the small trial size and short duration of study.

While PD1 is presented on the surface of T cells and can be accessed easily by anti-PD1 therapies, other non-surface regulators of T cell function are harder to be reached and exploited via antibody-based therapy. With the emergence of the CRISPR-Cas9 system, these internal regulators can be easily accessed to study for their role in enhanced anti-tumor T cell functions. One such internal regulator, the cytokine-inducible SH2-containing protein (CISH), has been knocked out using the CRISPR-Cas9 system and has been demonstrated to enhance reactivity towards cancer antigens in TIL therapy [58]. Two ongoing trials [59,60] are currently studying the use of TILs with disruptions in CISH for non-small cell lung and gastrointestinal cancers.

**Table 1 cells-12-01103-t001:** List of autologous cellular immunotherapies using the CRISPR-Cas9-based gene editing system.

Conditions Targeted	Targets Knocked out via CRISPR	Targets Knocked in (via Lentivirus or CRISPR)	Sponsor	Clinical Trial ID
Advanced hepatocellular carcinoma	PD1		Central South University	NCT04417764 [43]
Advanced esophageal squamous cell carcinoma	PD1		Hangzhou Cancer Hospital	NCT03081715 [44]
Metastatic gastrointestinal cancers	CISH		Intima Bioscience, Inc.	NCT04426669 [60]
Metastatic non-small cell lung cancer				NCT05566223 [59]
Metastatic non-small cell lung cancer	PD1		Sichuan University	NCT02793856 [45]
EBV^+^ malignancies	PD1		Nanjing University	NCT03044743 [46]
CD5^+^ relapsed/refractory T cell malignancies	CD5	CD5-CAR (via lentivirus)	Huazhong University	NCT04767308 [61]
Acute lymphocytic leukemia	HPK-1	CD19-CAR (via lentivirus)	Xijing Hospital	NCT04037566 [62]
Multiple solid tumors	PD1 and TRAC	Mesothelin-CAR (via lentivirus)	Chinese PLA General Hospital	NCT03545815 [47]
Mesothelin-positive multiple solid tumors				NCT03747965 [48]
Advanced EGFR-positive solid tumors	TGF-β receptor Ⅱ	EGFR-CAR (via lentivirus)		NCT04976218 [63]
Multiple myeloma	PD1, TRAC and TRBC	NY-ESO-1-TCR (via lentivirus)	University of Pennsylvania	NCT03399448 [49]
Acute myeloid leukemia	TRBC and TRAC	Wilms Tumor 1-TCR (via CRISPR)	Intellia Therapeutics	NCT05066165 [64]

#### 2.1.3. Role of CRISPR-Cas9 Tools in Design of Allogenic ”Off-the-Shelf” T Cell Therapies

All the six CAR-T therapies approved so far are autologous in nature with each patient-specific CAR-T cell needing to be manufactured separately, placing a heavy burden on the supply chain. Autologous therapies pose further challenges for late-stage cancer patients who have undergone several rounds of treatments, as their T cells cannot be scaled up to the desired dose levels [65]. Creating a stockpile of allogenic CAR-T cells would ease manufacturing challenges and give the late-stage patients a shot at this therapy. However, there are challenges in creating effective allogenic CAR-T therapies, including mismatches in the donor and the recipient’s human leukocyte antigens (HLA) leading to the onset of graft vs. host disease (GVHD) and rejection of the allogenic CAR-T cells by the host. Mechanistically, endogenous TCRs and major histocompatibility class molecules (MHC Classes I and II) are the mediators of self- and non-self-discrimination in T cells, and CRISPR-Cas9 tools have been used to disrupt these molecules to create allogenic therapies [65,66,67] (Figure 3). A number of registered clinical trials are focused towards disrupting the *TRAC* loci of the TCR [50,68,69,70,71,72,73,74] and β2-microglobulin gene [68,71,72,73,74], an MHC Class I molecule.

So far, phase 1 safety data [66] from one trial [70] has been published recently involving the use of CRISPR-Cas9-engineered allogenic CD19-specific CAR-T cells for the treatment of children with refractory B cell leukemia. In this study, CRISPR-Cas9 was used to target an endogenous TCR locus (*TRAC*) as well as the gene coding for CD52. Results from this trial indicated that allogenic CAR-T cells disappeared four weeks after infusion, and four out of six patients exhibited complete remission, indicating evidence of anti-tumor activity by the allogenic T cells. Two patients did not display any expansion of the therapeutic T cells leading to progression in disease. Given the allogenic nature of these cells, there is a high risk of GVHD, and hence, this study was performed under high doses of lymphodepleting drugs such as cyclophosphamide, and yet, two patients developed GVHD. Given the high doses of lymphodepleting drugs, there were eight instances of viral infections as well. The results from this trial highlighted some of the promises and challenges in designing an effective allogenic cancer CAR-T therapy.

Using the CRISPR-Cas9 system to disrupt one or more MHC class I and II molecules could reduce the risks of GVHD and mediate better tolerance of these cells by the host, eventually paving the way for these therapies to be performed at lower doses of lymphodepleting drugs with lower risks of viral infections. Another ongoing trial [50], performed by Caribou Biosciences, to treat relapsed or refractory B cell non-Hodgkin lymphoma involves an allogenic T cell product (CB-010), where healthy donor T cells are taken, and the endogenous TCR was disrupted at the *TRAC* loci. To extend the anti-tumor activity of these cells, *PDCD1* was interrupted as well. In addition, an CD19-specific CAR cassette was precisely inserted at the *TRAC* loci using the CRISPR-Cas9 system to regulate the expression of CAR [75]. The preliminary results [76] from this study indicated that all six patients who had received this treatment showed a complete response (CR) to the treatment. After 6 months, three of the six patients maintained CR; after 12 months, two of six patients maintained CR; and after 18 months, one patient maintained CR. More details on side effects such as GVHD, viral infections, etc., are unavailable at the time of writing of this paper. In summary, CRISPR-Cas9 tools have provided the flexibility of combining many strategies such as disruptions of PD1 and other immune check point regulators along with the removal of endogenous TCR and MHC class molecules to create a T cell therapeutic that is both safe and enduring as well as more sustainable from a manufacturing and cost perspective.

**Table 2 cells-12-01103-t002:** List of allogenic cellular immunotherapies using the CRISPR-Cas9-based gene editing system.

Conditions Targeted	Targets Knocked out via CRISPR	Targets Knocked in (via Lentivirus or CRISPR)	Sponsor	Clinical Trial ID
Acute myeloid leukemia	CD33	None	Vor Biopharma	NCT05309733 [77]
Relapsed or refractory CD19^+^ leukemia and lymphoma	TRAC and β_2_M	CD19 CAR (via lentivirus)	Chinese PLA General Hospital	NCT03166878 [68]
Relapsed or refractory leukemia and lymphoma		CD19^+^ CD20 CAR or CD19^+^ CD22 CAR (via lentivirus)		NCT03398967 [69]
B cell acute lymphoblastic leukemia	TRAC and CD52	CD19 CAR (via lentivirus)	Great Ormond Street Hospital	NCT04557436 [70]
Elapsed/refractory B cell non-Hodgkin lymphoma	TRAC and PD1	CD19 CAR at TRAC loci (via CRISPR)	Caribou Biosciences, Inc.	NCT04637763 [50]
Relapsed or refractory T or B cell malignancies	TRAC, β_2_M and CD70	CD70 CAR at TRAC loci (via CRISPR)	CRISPR Therapeutics AG	NCT04502446 [71]
Renal cell carcinoma				NCT04438083 [72]
B cell malignancy	TRAC and β_2_M	CD19 CAR at TRAC loci (via CRISPR)		NCT04035434 [73]
Relapsed or refractory multiple myeloma	TRAC and β_2_M	BCMA CAR at TRAC loci (via CRISPR)		NCT04244656 [74]

#### 2.1.4. Using CRISPR-Cas9 Tools for Precise Insertion of CARs in T Cell Therapies

CARs in CAR-T therapy as well as engineered TCRs in TCR-T therapy are inserted into T cells via lentivirus and other retroviral vectors. These viruses insert the engineered receptor cassettes randomly in the genome and often at multiple locations leading to mutations in non-desired genes and differential expression of these cassettes within the population [78]. Using the CRISPR-Cas9 system, precise insertion of CARs under promoters of endogenous TCR genes have enabled controlled expression of these cassettes and have been shown to delay T cell exhaustion and differentiation with improved anti-tumor activity [75]. In recently launched clinical trials [50,64,71,72,73,74], CRISPR-Cas9 has been used instead of lentivirus to specifically knock-in CARs under the *TRAC* loci of TCR to control the CAR expression. The interim results from one such trial conducted by Caribou Biosciences involving the product CB-010 are discussed in the previous section. This trial and several others are still ongoing, and complete results are yet to be published.

#### 2.1.5. Role of CRISPR-Cas9 Tools to Mediate Effective CAR-T Therapy towards T Cell Malignancies

CAR-T therapies have been more successful against B cell-related malignancies than T cell-related malignancies. One of the reasons for CAR-T being ineffective against T cell-related cancers is due to the fratricide effects mediated by CAR-T cells where these cells target their fellow CAR-T cells carrying the same antigens that they are engineered to recognize and fight against. One trial [61] is focusing on removing CD5 in CAR-T cells to make them effective against T cell malignancies. While there is a baseline level of expression of CD5 in thymocytes, T, and B1 cells, CD5 expression is elevated in over 85% of T cell-related malignancies. Engineering a CAR-T therapy against CD5 molecule to target T cell malignancies might lead to fratricide due to the presence of the same CD5 molecule in engineered CAR-T cells as well. In preclinical studies [79], CRISPR-Cas9 tools have helped in mitigating fratricide effects by selectively removing such antigens in CAR-T cells and focusing the anti-cancer activity towards cancer cells.

In addition to fratricide effects, engineering an autologous T cell therapy for T cell malignancies risks injecting a lymphoblastic T cell into the patients. To overcome such risks, an allogenic T cell therapy from healthy donors can negate the risk of injection of lymphoblastic T cells in patients. Two ongoing trials [71,72] conducted by CRISPR Therapeutics are focused on evaluating an allogenic CAR therapy (referred as CTX130™) for treating CD70^+^ relapsed or refractory T or B cell malignancies and renal cell carcinomas. Here, T cells are taken from healthy donors, CD70 loci and the two loci for TCR (*TRAC* and *TRBC*) are disrupted using the CRISPR-Cas9 system, and a CD70-specific CAR is precisely inserted into *TRAC* loci using the CRISPR-Cas9 system. Initial results indicated that in patients who had high doses (>300 million cells), there was a complete response in 29% of patients, and progression of disease was prevented in 100% of patients. There were no side effects such as cytokine release syndrome (grade 3 or higher) in these high dose group of patients in the trial [80]. This trial is still ongoing, and complete results are yet to be published.

#### 2.1.6. CRISPR-Cas9 Tools to Reduce Side Effects of CAR-T-Based Therapies

CRISPR-Cas9 systems have also helped in studying ways to mitigate other side effects of CAR-T therapies including cytokine release syndrome (CRS), prevalent in adult B cell acute lymphoblastic leukemia (BLL) patients when treated with the CD19-specific CAR-T cell therapy. Pre-clinical studies have indicated that CRISPR-Cas9-mediated decreased expression of HPK-1 in CAR-T cells mitigated the CRS effects while maintaining an effective anti-tumor capability [81]. These findings are being validated in a phase 1 clinical trial [62] in patients and results indicated that 72.7% of patients (8 out of 11) had complete remissions of cancer with no patient experiencing CRS (grade 3 or higher) or neurological events [82].

#### 2.1.7. Role of CRISPR-Cas9 Tools in TCR Therapies

Beyond the CAR-T- and TIL-based therapy described above, adoptive cell therapies also include therapies based on T cells with engineered TCRs. In this approach, exogenous TCRs that are designed to target a specific antigen are engineered into T cells. To prevent the mispairing and/or competing of exogenous TCRs with endogenous TCRs, both the α and β chains of the endogenous receptor are removed by knocking out *TRAC* and *TRBC* using CRISPR-Cas9 tools. Two clinical trials have used this approach to target multiple myeloma [49] and acute myeloid leukemia [64]. The NCT03399448 [49] trial targeting multiple myeloma [49] is one of the first trials to publish data on safety and efficacy of CRISPR-Cas9-engineered T cells where, in addition to removing α and β chains of the endogenous TCR, the immune checkpoint regulator PD1 was also removed to enhance the anti-tumor activity of these cells with engineered TCRs [51].

Recently, Intellia Therapeutics initiated a trial [64] for evaluating their autologous TCR therapy for treatment of acute myeloid leukemia. Their product, named NTLA-5001, comprised T cells that had their *TRBC* and *TRAC* loci disrupted sequentially using CRISPR-Cas9, and later, an exogenous TCR specific to Wilms Tumor 1-specific antigen was inserted into the *TRAC* loci using CRISPR-Cas9 [83]. As of October 2022, Intellia therapeutics has indicated that this trial has been stopped, and a similar allogenic product is under preclinical development. In future, CRISPR-Cas9 tools are expected to play a significant role in the advancement of engineered TCR-based therapies as well.

#### 2.1.8. Gene Editing Efficiencies and Safety Profiles of CRISPR-Cas9-Edited T Cell Therapies

In the multiple clinical studies (NCT03399448 [49], NCT03747965 [48], NCT02793856 [45], NCT04557436 [70]) that have been published so far on CRISPR-Cas9-mediated gene editing in T cell therapies, gene-edited products were found to be clinically acceptable and safe. One of the major risks associated with CRISPR-Cas9-mediated gene editing has been the off-target editing. While all four studies cited above identified off-target edits, their frequencies were low between 0.05–4% in the overall gene-edited populations. The first published trial involving CRISPR-Cas9 editing in T cell therapies (NCT03399448 [49]) involved simultaneous editing of three different genes (*TRAC*, *TRBC*, and *PDCD1*) via electroporation of Cas9 with three different sgRNAs, one for each target. As a result, the cell population had *TRAC* edited in only 45% of cells, *TRBC* in 15%, and *PDCD1* in 20% of cells. More importantly, only 10% of cells had edits at all the three loci, with over 30% of cells having no edits in any of the loci. These statistics indicate that the overall low editing efficiency of the CRISPR-Cas9 system used in this study. Though there were off-target edits for each of the locus, the vast majority of the cells had on-target edits, with average on-target editing frequency across three patients in the trial was 99.4% for the *PDCD1* locus, 98.6% for the *TRAC* locus, and 95.% for the *TRBC* locus. The risk of translocation exists during simultaneous editing of multiple loci, and translocations were observed in all manufactured products; however, the translocation frequencies were low and close to the limits of detection of the qPCR assay used for measurement. In general, translocations decreased over time during manufacturing as well as post infusion, indicating no positive selection for these translocations. Subsequently, after four months of infusion, barring one large 9.3 kb deletion, all other translocations were not detected [51].

Another study [52] used electroporated plasmids containing genes for Cas9 and sgRNAs to edit *PDCD1* locus in T cells and observed a median editing efficiency of 5.8% in cell products across 12 patients. Off-target effects were measured at 18 different potential off-target sites, and the median off-target frequency was 0.05% across all patients with the majority of these effects occurring in the intergenic and intronic regions. In another trial [48], published by Han et al. [53], CRISPR-Cas9 was used to simultaneously target two genes, *TRAC* and *PDCD1*, via electroporation of Cas9 and two sgRNAs and reported a mean editing efficiency of 87.6% in the *PDCD1* locus and 95.7% in the *TRAC* locus. To measure possible off-target effects, 26 different potential off-target sites were genotyped, and no off-target edits were observed. Since two sites were edited simultaneously, translocation frequencies were measured as well and were found to be similar to previous CRISPR-based [51] and TALEN-based multi-gene editing studies [84].

Interestingly, Intellia Therapeutics unveiled their gene editing approach [83] for their NTLA-5001 product evaluated in a clinical trial [64]. Here, they employed a sequential editing approach to KO first the *TRBC* locus of TCR receptor using a lipid nanoparticle (LNP) containing Cas9 and a sgRNA. Later, in a subsequent round of gene editing, the *TRAC* locus was targeted followed by the precise insertion of CAR at the *TRAC* locus. The DNA template coding for engineered TCR was delivered via an AAV6 virus instead of electroporation or lipid-based transfection of a double-stranded DNA. Sequential gene editing at different targets avoided risks of translocation, and they reported endogenous TCR KOs in 98% of cells with precise exogenous TCR insertion at the *TRAC* locus in over 50% of cells. This clinical trial was initially designed to be an autologous therapy, but later, it was discontinued and is now under development to be an allogenic product. Currently, the impediment in assessing the safety of these products is that in studies that are completed so far, the size of the trials are small and have employed products with low gene editing efficiencies. As the editing efficiency improves, a fresh set of assessments for safety and efficacy of various approaches described above would be required. The outcome of many of ongoing clinical trials may provide the future directions for this field.

### 2.2. CRISPR-Cas9 Therapies for Other Non-Infectious Diseases

Information related to these subsequent clinical trials can be found in Table 3.

#### 2.2.1. Cell Therapy for Sickle Cell Disease (SCD) or Transfusion-Dependent β-Thalassemia (TDT)

Sickle cell anemia is a genetic disease caused by homozygous mutations in the β-globin gene, resulting in misshaped and rigid red blood cells that are prone to clog small blood vessels and deprive tissue of oxygen (vaso-occlusive crisis); further, in sickle cell disease, red blood cells are fragile and die easily, leading to a shortage of red blood cells (anemia). Similarly, β-thalassemia is a genetic blood disease in which the body fails to produce adequate hemoglobin due to deleterious mutations in β-globin resulting in anemia; patients with the most severe disease require regular blood transfusions to survive (transfusion-dependent β-thalassemia). In both diseases, increased expression of fetal hemoglobin over adult hemoglobin is associated with better patient outcomes. Several groups are developing treatments that have demonstrated promise as a functional cure for one or both diseases and are described below.

A collaboration between CRISPR Therapeutics and Vertex Pharmaceuticals produced the CRISPR-Cas9 therapeutic exagamglogene autotemcel (“exa-cel”, formerly CTX001) to functionally cure both sickle cell disease (SCD) and transfusion-dependent β-thalassemia (TDT) [85,86,87,88,89,90]. In this therapy, CD34^+^ hematopoietic stem and progenitor cells (HSPCs) are isolated from a patient, modified with exa-cel to knock-out the expression of a repressor to fetal hemoglobin (*BCL11A*), thereby increasing expression of fetal hemoglobin in the blood to compensate for the loss of adult hemoglobin (refer to Khosravi et al. [91] for a clear illustration of how BCL11A facilitates the switch from fetal to adult hemoglobin). The edited cells are then returned to the patient. The therapeutic goal is for patients to require fewer blood transfusions (β-thalassemia) and experience fewer vaso-occlusive crises (sickle cell disease). Early study results on the first two patients to receive exa-cel reported the editing efficiency of the exa-cel therapeutic to be 68.9% for patient one (received exa-cel for β-thalassemia), and for patient two, there was 82.6 and 78.7% editing efficiency (patient two received two lots of exa-cel for sickle cell disease) [92]. Subsequent early study results with 31 patients with sickle cell disease reported that after a single exa-cell infusion, all patients in the trial experienced no adverse events related to therapy and no vaso-occlusive crises for 2–32 months after exa-cell therapy (for reference, patients experienced 2–9.5 severe vaso-occlusive crises in the two years preceding exa-cel) [93]; at six months post infusion, the average percentage of bone marrow CD34^+^ HSPCs or peripheral blood mononuclear cells (PBMCs) with an edited *BCL11A* allele was 86.6% or 76.0%, respectively. Additionally, early results from a β-thalassemia trial indicated that out of the 44 individuals in the trial, 42 of them did not require RBC transfusion post therapy in the 1–36 months post exa-cel; the 2 patients that had not stopped RBC transfusion had reduced the transfusion volume by ~80% (for reference, these 44 individuals had received between 15 and 71 RBC transfusions in the two years preceding exa-cell therapy) [94]; editing efficiencies were somewhat reduced in this study compared to the aforementioned trial for sickle cell disease, with the average percentage of bone marrow CD34^+^ HSPCs or PBMCs containing an edited *BCL11A* allele was 74.3% or 63.4%, respectively

EdiGene and Bioray Laboratories used a similar approach as in exa-cel in their respective therapies, ET-01 and BRL-101; both therapies use CRISPR-Cas9-mediated KO of *BCL11A* expression in autologous CD34^+^ HSPCs in β-thalassemia patients [95,96,97]. Preliminary safety and efficacy results on a single patient were published for ET-01, demonstrating ~80% editing efficiency in the ET-01 therapeutic and after infusion, editing frequency at 52 weeks of ~60 and ~80% in peripheral blood and bone marrow, respectively [98]; the patient saw in increase in fetal hemoglobin and after 87 days post ET-01, the patient stopped requiring blood transfusions, staying transfusion-free for at least 15 months. Likewise, initial data from a study investigating BRL-101 in two children with β-thalassemia demonstrated engraftment of CRISPR-Cas9-modified HSPCs and increases in fetal hemoglobin expression, and both patients did not require a blood transfusion for >18 months post treatment [99]. This study demonstrated high editing rates in the BRL-101 therapeutic (97.17 and 98.22% for patient 1 and 2, respectively), and post therapy, editing efficiencies of PBMCs were in the range of 61.72–79.55% over an 18-month period.

**Table 3 cells-12-01103-t003:** List of CRISPR-Cas9-based gene editing therapies for other non-infectious diseases.

Conditions Targeted	Gene Target	Edit Type	Therapeutic	Sponsor	Clinical Trial ID
Sickle cell disease or β-thalassemia	BCL11A	KO (NHEJ)	exa-cel	CRISPR Therapeutics and Vertex Pharmaceuticals	NCT03655678 [85]
					NCT05477563 [86]
					NCT03745287 [87]
					NCT05356195 [88]
					NCT05329649 [89]
					NCT04208529 [90]
			ET-01	EdiGene (GuangZhou) Inc.	NCT04925206 [95]
					NCT04390971 [96]
			BRL-101	Bioray Laboratories	NCT05577312 [97]
	β-globin	HDR	nula-cel	Graphite Bio, Inc	NCT04819841 [100]
			CRISPR_SCD001	UCLA, UC Berkeley	NCT04774536 [101]
			iHSCs with corrected β-globin	ALLIFE Medical Science and Technology	NCT03728322 [102]
β-thalassemia	γ-globin promoter	KO (NHEJ)	BRL-101	Bioray Laboratories	NCT04211480 [103]
			EDIT-301	Editas Medicine, Inc.	NCT05444894 [104]
Sickle cell disease					NCT04853576 [105]
Type 1 diabeties	proprietary		VCTX210A	CRISPR Therapeutics and ViaCyte	NCT05210530 [106]
Leber congenital amaurosis 10	CEP290		EDIT-101	Editas Medicine, Inc.	NCT03872479 [107]
Hereditary angioedema	KLKB1 (liver)		NTLA-2002	Intellia Therapeutics	NCT05120830 [108]
Duchenne muscular dystrophy	Dp427c	Exon skipping	CRD-TMH-001	Cure Rare Diseases, Inc	NCT05514249 [109]

Several other companies are also developing therapies using CRISPR-Cas9 gene editing therapies for TDT and SCD. Editas Medicine is testing EDIT-301, an autologous cell therapy comprising CD34^+^ hematopoietic stem cells edited to promote fetal hemoglobin expression (NCT05444894 and NCT04853576) [104,105]. It was previously observed that some mutations within the distal CCAAT-box region of fetal hemoglobin (**γ**-globin) promoters resulted in persistence of fetal hemoglobin in adults; EDIT-301 mimics naturally occurring mutations through Cas12a RNP-mediated disruption of this region to promote the expression of gamma globin (and thus the formation of fetal hemoglobin) [110,111]. Early clinical data from the first two patients that received EDIT-301 for sickle cell disease had positive initial findings with successful hematopoietic engraftment and an increase in fetal hemoglobin levels; further, both patients did not have vaso-occlusive crises in the 1.5 or 5 months following treatment [112].

Finally, two separate teams are using homology-directed repair to replace the SNP that causes sickle cell disease. Graphite Bio is evaluating their drug nulabeglogene autogedtemcel (nula-cel, formerly GPH101), which uses a high-fidelity Cas9 and non-integrating template DNA delivered from AAV6 to correct the β-globin gene through homology-directed repair (HDR) in autologous CD34^+^ HSPCs cells [113]. A press release from Graphite Bio indicated the first patient had received nula-cel [114], though the trial has subsequently and voluntarily been paused pending an adverse event that occurred in the first patient [115]. UCLA and UC Berkley researchers developed CRISPR-SCD001, which similarly corrects the β-globin gene through HDR, though in this case, the CRISPR-Cas9 system is delivered ex vivo via electroporation [101]. No additional information on this study was available to date.

#### 2.2.2. Type 1 Diabetes

A few years ago, ViaCyte published early clinical results demonstrating successful implantation of a visualizable device containing pancreatic progenitor cells differentiated from a proprietary human pluripotent stem cell line (CyT49) for the purposes of treating type 1 diabetes [116,117]. These studies showed that implanted pancreatic progenitor cells survived and matured into beta cells which produced insulin in a meal-responsive manner. However, while promising, patients in the study had to take immunosuppressive drugs to avoid allogenic cell rejection, and several adverse events were observed in the study that were attributed to immunosuppression itself.

In an effort to avoid allogenic cell rejection while providing a functional cure for type 1 diabetes, ViaCyte joined efforts with CRISPR Therapeutics to create a combination therapy called VCTX_210_, which comprised the same transplantable device containing pancreatic beta-cell precursors derived from CyT49 cells transgenically modified to prevent triggering an immune reaction using a proprietary CRISPR-Cas9 scheme [106]. The first patient was transplanted with VCTX210 in February 2022 [118], though no preliminary results updates were found to date.

#### 2.2.3. Leber Congenital Amaurosis 10 (LCA10)

Editas Medicine developed EDIT-101 for the treatment of Leber congenital amaurosis (LCA10), a severe retinal disease that leads to blindness that is caused by mutations in the *CEP290* gene, the most common of which is an intronic A-to-G mutation causing a splice donor site and corresponding cryptic exon insertion [119]. EDIT-101 is a subretinal injection of AAV5 containing DNA for Cas9 and two guide RNAs; the gRNAs target two regions flanking the pathogenic intronic mutation in *CEP290* (IVS26), excising or inverting it to restore function [120]. The results from 14 patients treated with EDIT-101 indicated 3/14 patients responded to treatment with improvements in retinal sensitivity and/or visual tests; of these, 2 patients were identified as homozygous for the IVS26 mutation [121]; subsequently, Editas has paused the clinical trial in search of a collaborative partner to work with on this therapy.

#### 2.2.4. Hereditary Angioedema

Hereditary angioedema (HAE) is a genetic disorder caused by an autosomal dominant mutation in *KLKB1*, encoding for prekallikrein, which is produced in the liver and circulates in the blood, where it is converted to kallikrein by the protease Factor XII. It is characterized by severe and reoccurring swelling (HAE attacks) in various regions of the body, causing pain or threatening life (e.g., in the case of throat swelling). Intellia Therapeutics developed NTLA-2002, an in vivo therapy for hereditary angioedema, which is delivered intravenously and comprises LNPs containing Cas9 mRNA and sgRNA specific to *KLKB1* [108]. Upon intravenous delivery, NTLA-2002 is taken up by liver cells where it causes indels in *KLKB1*, knocking out function [122]. The interim clinical study results for 10 patients receiving NTLA-2002 reported that the therapeutic was well tolerated by patients, who experienced a dose-dependent reduction in plasma kallikrein and large reduction in HAE attacks over the study period, with the first three patients being attack free for 5–10 months post treatment (for reference, this patient cohort suffered about 1–7 attacks per month prior to treatment) [123].

#### 2.2.5. Duchenne Muscular Dystrophy

As of 15 December 2022, the ClinicalTrials.gov database contains only one clinical study involving the use of a CRISPR therapeutic intervention aimed at altering gene expression (NCT05514249, Table 4) [109]. The phase 1 study, sponsored by Cure Rare Diseases, Inc in collaboration with UMass Chan Medical School, enrolled a single patient with Duchenne muscular dystrophy (DMD) who was treated with a single intravenous dose of the investigational new drug (IND) CRD-TMH-001. DMD is a rare, X-linked recessive disorder that affects ~1 in 3500 males worldwide [124]. In DMD, most affected individuals lack the dystrophin protein due to mutations that result in frameshift errors. The dystrophin gene spans 2.6 Mb and contains 79 exons and is the largest gene in the human genome. The Cure Rare Diseases therapeutic relied on CRISPR-mediated exon skipping, targeting one or more exons that were causing frameshift errors to produce an upregulated isoform of dystrophin. Exon skipping typically involves targeting antisense oligonucleotides to introns that flank the exon of interest in pre-mRNA to induce splicing of the aberrant exon. Similarly, CRISPR-Cas can be used to induce alternative splicing. Cure Rare Diseases describes using CRISPR components regulated by a muscle-specific promoter, packaged in a patient-matched AAV serotype with low immunogenicity, that targets delivery to muscle cells.

### 2.3. CRISPR-Cas9 Therapies for Viral Infections

#### 2.3.1. Human Immunodeficiency Virus (HIV-1)

Excision BioTheraputics evaluated dosages and long-term safety of the in vivo biologic EBT-101 for the treatment of HIV-1 infection [125,126]. EBT-101 is an AAV9-delivered CRISPR-Cas9 therapeutic given intravenously to target and excise HIV-1 proviral DNA (the viral form that has integrated into the genome of infected cells and is impervious to standard treatments). EBT-101 gRNAs target HIV-1 proviral DNA and create multiple dsDNA breaks to cut out a large section of the proviral genome, which, when repaired through NHEJ, renders the HIV-1 proviral genome incomplete and unable to replicate, effectively removing it from reservoir cells and tissues. In vivo preclinical studies targeting simian immunodeficiency virus (SIV) in rhesus macaques demonstrated the therapy was well tolerated, with successful excision of HIV-1 proviral DNA from a wide-range of tissue reservoirs, including CD4^+^ T cells, brain, and lymph nodes, among others [127]. Recently, the first human patient was dosed with EBT-101 (July 2022), with the first press release reporting the treatment had been well tolerated and the clinical trial was moving forward as planned [128].

Ex vivo cell therapy approaches have also been proposed for the prevention or treatment of HIV. One clinical trial from Affiliated Hospital to Academy of Military Medical Sciences in Beijing aims to investigate the safety of transplanted allogenic CD4^+^ cells that have had KO of CCR5, a cell coreceptor for HIV entry [129]; indeed, non-functional CCR5 alleles have been observed in naturally HIV-resistant humans [130]. A case report was published for a single patient who was diagnosed with an HIV infection and, shortly thereafter, T cell acute lymphoblastic leukemia (T-ALL) [131]. Following antiviral therapy that resulted in undetectable virus in serum and chemotherapy for T-ALL, the patient received allogenic hematopoietic stem cell transplantation. Prior to transplantation, CD34^+^ HSPCs were isolated and edited to KO the CCR5 gene with the CRISPR-Cas9 approach. Transplanted cells were successfully engrafted, with cells with a modified CCR5 gene persisting for at least 19 months. However, post therapy, the proportion of circulating bone marrow cells that had a disrupted CCR5 gene was low (<8%), which was insufficient to cure HIV-1 infection, highlighting that improvements in CRISPR-Cas9 editing efficiency in this system are needed.

#### 2.3.2. Human Papilloma Virus (HPV)

A clinical trial sponsored by First Affiliated Hospital at Sun Yat-Sen University in Guangzhou, China proposes to use two in vivo approaches to treat persistent human HPV infection and related cervical malignant neoplasia: a TALEN-based and a CRISPR-Cas9-based therapy [132]. HPV, once integrated into the host genome, is particularly difficult to treat. Each therapy would be encoded in a plasmid administered in a gel and would target two HPV genes (the E6 and E7 oncogenes) to destroy the integrated viral genome, causing apoptosis or growth inhibition of infected cells. The status of this study is unknown and was last updated in 2017. In the interim, the team has been publishing results from their efforts to improve editing efficiency of the HPV E6/E7 oncogenes through a method called gene knock-out chain reaction (PMID: 35036522) [133], and exploration of a CRISPR/Cas13a system to knock-out HPV E6/E7 mRNAs in vivo cell culture [134].

**Table 4 cells-12-01103-t004:** List of CRISPR-Cas-based gene editing therapies for infectious diseases.

Condition Targeted	Gene Target	Edit Type	Therapeutic	Sponsor	Clinical Trial ID
HIV-1	HIV proviral DNA	Viral genome split (NHEJ)	EBT-101	Excision Biotherapeutics	NCT05144386 [125]
					NCT05143307 [126]
HPV	E6/E7 genes of HPV16/18	Viral genome split (NHEJ)	Talen: TALEN-HPV16 E6/E7 or TALEN-HPV18 E6/E7; CRISPR-Cas9: CRISPR/Cas9-HPV16 E6/E7T1 or CRISPR/Cas9-HPV18 E6/E7T2	First Affiliated Hospital, Sun Yat-sen University	NCT03057912 [132]
Viral keratitis	HSV-1 genome	Viral genome split (NHEJ)	CRISPR/Cas9 mRNA	Shanghai BDgene Co., Ltd.	NCT04560790 [135]

#### 2.3.3. Viral Keratitis, Herpes Simplex Virus 1 (HSV-1)

BDgene Co investigated safety and dosing of an in vivo CRISPR-Cas9 mRNA therapy (BD111) for corneal inflammation (keratitis) caused by HSV-1 infection [135]. BD111 is delivered by lentivirus into the cornea. In preclinical research, this therapy demonstrated transport from the cornea into neurons (trigeminal ganglia, which are viral reservoirs) with a short life span of Cas9 activity (three days post injection); following injection, viral replication is inhibited by guiding CRISPR-Cas9 activity to specific regions of the HSV-1 genome, causing a large genome excision subsequent INDEL formation [136]. Clinical investigation of BD111 is primed to move forward having received FDA orphan drug designation in June 2022 [137].

## 3. Base Editing

### 3.1. Principles of Base Editing

One of the most promising advancements utilizing the CRISPR system is that of base editing. Developed by David Liu’s lab in 2016 [138], the first base-editing-based therapeutics made their way into phase I clinical trials in 2022, with many more in the pipeline. Base editing works by directly converting one or more DNA bases to another (for example, C to T) without the initiation of a double-strand break or the use of repair templates. Early efforts in base editing used either an inactive Cas9 or a Cas9 nickase fused to a cytidine deaminase to directly convert cytidine to uridine, resulting in a C-to-T substitution [138,139]. Thus, the Cas9 effectively serves as a programmable guide to direct the deaminase to the desired chromosomal location. When the guide RNA binds to the target DNA sequence and forms an RNA:DNA hybrid, this leaves a stretch of single-stranded DNA (ssDNA) directly adjacent to the PAM sequence. Since ssDNA is the substrate for deaminase enzymes, the ssDNA in this bubble can then be targeted by the base editor. The nucleotides closest to the PAM are obstructed by the Cas9 and are unable to be edited, but the nucleotides furthest from the PAM are accessible to the deaminase, facilitating base editing in a narrow window at a set distance from the PAM sequence. Since the initial invention of base editing, much work has been conducted to improve the fidelity of the approach and improve the scope of the base conversions that are possible [11,140]. Using classical base editing, it is possible to make four nucleotide conversions (C to T, A to G, T to C, G to A) that represent almost one-third of pathogenic mutations in humans. A more recent advance, prime editing [141], can make the remaining nucleotide conversions and will be discussed in more detail later.

### 3.2. Clinical Applications of Base Editing

Since the invention of base editing in 2016, several companies have formed around base editing technology, with various therapeutics in the pipeline. In general, base editing therapeutics can be divided into the type of delivery method that is used (Figure 4). Some therapies are delivered directly to patients using either LNPs or a viral vector, while others are first delivered ex vivo to hematopoietic stem cells (HSCs) or T cells, typically through electroporation, and the edited cells are then reintroduced to the patient. Current base editors that have entered clinical trials are discussed below.

#### 3.2.1. In Vivo Delivery

An attractive feature of base editing is that it raises the potential to directly modify cells in tissues of patients with high efficiency and low probability of off-target effects. The first in vivo-delivered base editor to enter phase I clinical trials is sponsored by Verve Therapeutics and uses LNPs for delivery of the base editing machinery [142]. This therapeutic, named VERVE-101, is designed to treat heterozygous familial hypercholesterolemia (HeFH). HeFH is a commonly inherited genetic disorder affecting around 1 in 250 people globally and is caused by mutations in the low-density lipoprotein receptor gene (*LDLR*) that result in abnormally high LDL levels [143]. Rare forms of familial hypercholesterolemia can also be caused by gain-of-function mutations in *PCSK9*, which is preferentially expressed in the liver [144]. Naturally occurring *PCSK9* loss-of-function variants are also common (2–3% in some populations) and result in lowered LDL levels [145,146], suggesting *PCSK9* as a target for treating HeFH. The siRNA-based therapeutic inclisiran inhibits *PCSK9* expression and lowers LDL levels for several months [147], raising the possibility that an irreversible editing of *PCSK9* might confer longer-term or permanent lowering of LDL levels. VERVE-101 uses an adenine base editor to target and effectively knock-out *PCSK9* through the induction of a single base edit. While the first patient in the VERVE-101 clinical trial was just enrolled in July 2022, and therefore, no data are available to assess the safety or efficacy. Verve and collaborators have conducted several preclinical studies in animal models that delivered promising results [148,149,150]. In a nonhuman primate model, the base editor was delivered to cynomolgus monkeys using an LNP-based delivery approach. The targeting of *PCSK9* was very efficient, with 70% of liver *PCSK9* edited at the highest dose, resulting in an 83% reduction in blood PCSK9 levels. As a result, LDL-C levels were reduced by up to 69%, with durable effects lasting at least through the duration of the study, which was over 1 year [148].

#### 3.2.2. Ex Vivo Delivery

Base editing also offers the ability to modify cells that have been extracted from patients, primarily HSCs or T cells, in cell therapy applications. There are two candidates for such applications that have advanced to clinical trials. The first to be delivered to patients, in May 2022, is a phase I trial sponsored by the Great Ormond Street Hospital for Children (NCT05397184 [22]). For this therapy, named BE CAR-7, T cells were collected from healthy donors to make allogeneic CAR-T cells for later infusion into patients with relapsed/refractory T cell acute lymphoid leukemia. After collecting the donor T cells, the base editor designed to modify three genes (*TRBC*, *CD52*, *CD7*) was delivered via electroporation, followed by delivery of a chimeric antigen receptor with a lentiviral vector [151,152]. While no peer-reviewed studies have been reported about the outcomes for this trial, news reports have hinted at promising results by reporting which showed that the first patient enrolled in this trial is leukemia free six months after she received the edited CAR-T cells and a second bone marrow transplant [153].

The second candidate is sponsored by Beam Therapeutics and is named BEAM-101 (NCT05456880). In November 2022, Beam successfully enrolled its first patient in its phase I/II clinical trial for BEAM-101, which is designed to treat severe SCD and β-thalassemia. Beam’s approach is modeled after individuals with natural genetic variants that cause continued expression of fetal hemoglobin, which can prevent or alleviate SCD symptoms [154]. Accordingly, BEAM-101 is designed to modify HSCs ex vivo with a base editor to activate the expression of fetal hemoglobin before transfusion of the HSCs back into the patient [155]. Beam also has a base editing-based therapeutic named BEAM-102 in pre-clinical development that similarly aims to treat SCD. In this case, however, Beam seeks to induce a base edit in HSCs that will revert the SNP that causes SCD to the normal hemoglobin sequence.

## 4. Prime Editing

As discussed previously, cytosine and adenine base editors are not able to make all the possible nucleotide changes. Therefore, while the potential applications of base editing are vast, they are also limited by the enzymes tethered to the Cas9. In 2019, Liu’s lab also developed a technique termed prime editing [141], which has overcome some of the limitations of base editing, expanding the potential scope of this class of CRISPR-based applications. As in base editing, prime editing covalently attaches an enzyme to a Cas9 nickase to target a desired genomic locus. However, instead of a deaminase, prime editing uses a modified reverse transcriptase (RT). In addition, prime editing uses a modified version of the guide RNA termed the prime editing guide RNA (pegRNA), consisting of a typical guide RNA that also contains an RT template sequence with the desired edit. The Cas9 nickase first makes a single strand cut, and the resulting 3′ hydroxyl group from the cleaved DNA serves as the primer to initiate reverse transcription of the template sequence. After reverse transcription, cellular repair processes then incorporate the newly synthesized DNA into the target site, and the permanent editing of both strands is accomplished by cellular DNA repair machinery. The result is that any nucleotide change can be induced with prime editing, and short indels can also be made without inducing double-strand breaks. Importantly, prime editing can also induce changes at distances further from the PAM than first-generation base editors, which should provide more flexibility in edits and lower reliance on the precise location of the PAM sequence. While no clinical trials based on prime editing are underway, the scope of DNA edits afforded by prime editing makes this a promising future modality. However, compared with base editing, prime editing is less efficient and induces more indels [11], so improvements are needed for prime editing to reach its full potential.

## 5. CRISPR and Gene Regulation

### 5.1. CRISPR Interference/CRISPR Activation (CRISPRi/CRISPRa)

CRISPR interference (CRISPRi) and CRISPR activation (CRISPRa) rely on the use of catalytically inactive dCas9, fused to effector domains that either repress or activate gene transcription (CRISPRi or CRISPRa, respectively) [156,157]. While dCas9 lacks endonuclease activity, it retains the ability to interact with guide RNA and bind target DNA loci, functioning as an RNA-guided DNA binding protein [12]. For CRISPRi in mammalian systems, dCas9 is fused with transcriptional repressor domains, most notably the Krüppel-associated box (KRAB) domain, one of the most potent transcriptional repressors in the human genome [12,156,158]. KRAB-based transcriptional repression relies on recruitment of KAP1 and on KAP1 recruitment of additional co-repressors (e.g., HP1, SETDB1, histone deacetylases) that propagate heterochromatin [156,159]. In CRISPRa, dCas9 is fused with transcriptional activator domains that promote gene expression. In eukaryotic cells, activator domains used include VP64 (four tandem repeats of Herpes simplex virus VP16 domain) fused with additional activation domains including Rta (Epstein–Barr virus R transactivator), p65 (subunit of NFκB), and HSF1 [156,157,160,161]. The multidomain construct VPR (fusion of VP64, p65, and Rta) is often used in mammalian systems to recruit and stabilize transcription factors and activate transcription [162].

### 5.2. CRISPR Epigenetic Editors

Catalytically inactive dCas9 can also be fused to catalytic epigenetic effector domains to alter DNA methylation and histone modifications to induce targeted gene silencing or activation. DNA methylation effectors successfully used for CRISPR-targeted gene silencing in human cells include the de novo methyltransferase DNMT3A and DNMT3L domains, and DNA demethylation domains successfully used for CRISPR-targeted gene activation include TET1 [163,164,165,166]. Histone modifiers have also been fused to dCas9 to induce targeted acetylation or methylation of H3K27, and methylation of H3K4, H3K9, and H3K79 in human cells [167,168]. Combinatorial fusions that incorporate both transcriptional and repressor epigenetic domains, for example, KRAB and DNMT3A, have demonstrated synergistic silencing [169].

### 5.3. Applications of CRISPR Gene Regulation in Models of Human Disease

#### 5.3.1. Retinitis Pigmentosa

Retinitis pigmentosa (RP) is a rare genetic disease that leads to progressive vision loss and affects ~1 in 4000 people globally [170]. In a mouse model of RP where mice carry a spontaneous mutation of the rod-phosphodiesterase gene (rd10 mice), a dCas9-KRAB system was targeted to the *Nrl* gene, which regulates rod versus cone photoreceptor determination through activation of *Nr2e3*, a transcription factor that represses transcription of multiple cone-specific genes [171,172]. Repression of *Nrl* by dCas9-KRAB resulted in reprogramming of rods into cone-like cells which are resistant to rod photoreceptor retinitis pigmentosa-specific mutations and prevented secondary cone loss [173,174]. In another study, Böhm et al. used dCas9-VPR to transactivate expression of a rhodopsin homolog from *Opn1nw* in a rhodopsin-deficient (Rho^+/−^) mouse model for RP [175]. The leading cause of RP is mutations in the *RHO* photoreceptor gene, which encodes for the most abundant protein in rod cells of the retina [176,177]. However, phototransduction molecules in rods and cones are encoded by several distinct but functionally equivalent genes [178]. Activation of *Opn1nw* in Rho^+/−^ mice using dCas9-VPR resulted in amelioration of retinal degeneration and improved retinal function [175].

#### 5.3.2. Facioscapulohumeral Muscular Dystrophy

Facioscapulohumeral muscular dystrophy (FSHD) is a rare genetic disease that affects ~1 in 20,000 males and females of all ages, and leads to progressive muscle degeneration in the face, shoulders, and upper arms [179]. FSHD is linked to contractions or loss of methylation of the D4Z4 macrosatellite repeat array at 4q35, which allows aberrant full-length DUX4 expression (DUX4-FL) in skeletal muscle leading to muscle atrophy [180,181]. Epigenetic dysregulation of the FSHD locus is proposed to also contribute to DUX-FL expression and pathogenesis, since FSHD onset, progression, and severity is highly variable. Himeda et al. used lentiviral delivery of dCas9-KRAB to the DUX4 promoter or exon 1 in FSHD myocytes derived from patient bicep muscle and demonstrated reduced expression of DUX4-fl to ~45% of endogenous levels in FSHD myocytes [182].

#### 5.3.3. Cancer

Altered gene expression is a hallmark of cancer, from down regulation of tumor suppressor genes (e.g., *PTEN*, *BRCA1*, *CDKN2A*, *RASSF1*, *HIC1*) to upregulation of oncogenes (e.g., *GRN*, *FHL2*, *CNKSR1*) [183,184,185,186,187,188,189,190]. For example, the loss of expression of tumor suppressor genes on chromosome 10, notably, the phosphate and tensin homolog (*PTEN*) tumor suppressor gene, is a common feature of numerous cancers that arises from somatic mutations, indels, and transcriptional and post-transcriptional alternations [191]. Epigenetic silencing through methylation of the *PTEN* promoter region has been reported in numerous cancers including melanoma, endometrial, breast, gastric, and colorectal cancers [192,193,194,195,196]. Using dCas9-VPR (VP64-p65-Rta), PTEN expression can be reactivated in melanoma cell lines, resulting in the repression of AKT, mTOR, and MAPK oncogenic pathways and increased sensitivity to B-Raf and P13K/mTOR inhibitors [197]. In primary breast myoepithelial cells, Saunderson et al. demonstrated that transient transfection with dCas9 DNMT3A-3L induced a state of hypermethylation leading to a p16 repression-driven hyper-proliferation, preventing senescence, and potentially leading to early tumorigenesis [198,199]. This study demonstrated that “hit-and-run” epigenetic alterations can induce heritable altered cellular processes.

#### 5.3.4. Imprinting Diseases

Epigenetic editors may also help rare imprinting diseases, changing methylation in imprinting control centers where imprinted genes are inappropriately silenced or overexpressed, to treat these and restore proper imprinted gene expression. Approximately 100 genes in the human genome have been identified to be imprinted, and 9 different imprinting diseases have been described [200,201]. They are typically found in gene clusters, regulated by imprinting control centers (ICRs) that are either methylated (inactive) or unmethylated (active) under coordinated epigenetic control. During maternal oogenesis, ICRs typically become hypermethylated, while most ICRs in male germ cells are fully unmethylated. The ICRs control expression of long antisense transcripts that silence expression of the protein coding genes they overlap [202]. The E6-AP ubiquitin ligase expressed from *UBE3A* gene is imprinted in neurons, requiring a functional maternal copy for expression [203]. The paternal copy is silenced by a long non-coding RNA (*UBE3A-ATS*) of which the promotor/exon 1 region completely lacks methylation (maternal copy is fully CpG methylated). Angelman syndrome is a rare neurological disorder caused by deficiency in UBE3A expression, most often due to a 5- to 7 Mb maternal deletion in chromosomal region 15q11.2-q13, uniparental disomy, or translocation in the maternal copy [204]. Wolter et al. demonstrated that targeting sgRNA-Cas9 to Snord115 (a small nucleolar RNA gene located in the 3′ region of the *UBE3A-ATS*) resulted in the activation of paternal UBE3A in cultured human and mouse neurons. Using this same strategy, they used AAV to target neurons in a mouse model of AS during embryonic and early postnatal stages, which resulted in the expression of paternal UBE3A for at least 17 months and rescue of normal anatomical and behavioral phenotypes [205]. Approaches using dCas epigenome editors targeted to methylate CpG islands at the *UBE3A-ATS* transcriptional start site have also been proposed as a means of activating UBE3A expression [206].

#### 5.3.5. HIV

The major barrier to curing HIV-1 infection is viral latency [207]. One approach for a cure is to administer latency-reversing agents to purge cells harboring harboring latent viral genomes, then treat with antiretroviral therapy. Another approach is to promote a permanent latent state by inhibiting HIV-1 transcription factors. In line with the second approach, da Costa et al. used sgRNAs targeting the HIV-1 proviral genome (LTR1-LTR5) and CRISPR/dCas9-KRAB to block HIV-1 reactivation in latently infected T cells and myeloid cells treated with latency-reversing agents such as PKC agonists and HDAC inhibitors [208]. They found that targeting the LTR enhancer region provided the best repression, and one of their CRISPR constructs inhibited latency reactivation by 160×.

### 5.4. Challenges and Future Perspectives

Of the clinical trials listed at the ClinicalTrials.gov database as of 15 December 2022, there are no published studies relying on use of therapeutics using CRISPRi, CRISPRa, or CRISPR-epigenetic editing. These newer CRISPR technologies offer numerous benefits, most notably, less concern for introducing off-target permanent genomic alterations with associated error-prone repair [10]. However, many unknowns remain, which include unknown off-target events and their long-term effects, dosing requirements, and duration of heritability. Even though dCas9 is catalytically inactive, off-target effects can be problematic for certain genomic loci and can lead not only to genome-wide changes in gene expression, but also unintended indel mutations [209,210,211]. To optimize specificity, important considerations in construct design include: sgRNA that incorporates structural elements in addition to sequence specificity, precise positioning of sgRNA relative to transcription start sites, and incorporation of distal regulatory elements (e.g., enhancers and locus control regions) [212]. Additionally, the use of engineered dCas9 or Cas9 orthologues that rely on rarer PAM sequences have been proposed as alternatives [213]. Lastly, in native tissues, the gold standard for gene delivery is the use of recombinant adeno-associated viral (rAAV) vectors; however, dCas9 fusions (e.g., 5.8 kb dCas9-VPR) often exceed AAV genome packaging capacity (~5 kb) [214]. Optimized strategies for efficient and targeted payload delivery have used split rAAV systems or lentivirus vectors, but challenges with serotype matching and tissue tropism remain [215,216]. Directed evolution of viral vectors, the use of functionalized LPNs, or the use of viral-like particles (VLPs) for packaging and delivery are other viable options [217,218,219].

Using sgRNA-dCas systems to alter gene expression has vast potential in therapeutic development for numerous human diseases, especially those that may require a modularized precision strategy that can address multiple pathogenic features as in the case of cancer. For example, using multiplexed sgRNAs, gene expression patterns contributing to cell proliferation, evasion of growth inhibition, apoptosis escape, immune evasion, drug resistance, and sustained angiogenesis could be all be addressed in a single in vivo cancer therapeutic [212,220]. In instances where transient ectopic expression is sufficient in achieving the therapeutic goal, where constitutive dCas9 expression is not required, CRISPRi/CRISPRa approaches may serve as important tools. iPSC differentiation or cellular reprogramming to restore pluripotency are potential ex vivo application targets for CRISPRi/CRISPRa, and potential in vivo applications include reversal of retrovirus latency allowing for subsequent anti-retroviral treatment [162,221].

CRISPR-epigenetic editing provides additional important functionality for gene regulation-based therapeutics—inducing changes in DNA methylation and/or histone modification that are heritable and providing stable transcriptional changes over hundreds of cell divisions [222]. Importantly, diseases caused by aberrant epigenetic programming (e.g., functional allele is available on the other parent-of-origin chromosome but silenced, improper biallelic expression) such as rare imprinting diseases can be directly impacted by epigenetic editing, with curative potential if administered at appropriate developmental timing [206]. Other important applications of CRISPR-epigenomic editing include the ability to transactivate functionally equivalent genes (with or without combined disruption of a pathogenic allele) in diseases caused by a heterozygous dominant haploinsufficiency. To this end, there are three biotech companies working on CRISPR-epigenomic editing that launched this year: Chroma Medicine, founded by David Lui; Tune Therapeutics, co-founded by Charlei Gerlash and Fyodor Urnov; and Epic Bio, led by Amber Salzman, based on Stanley Qi’s research. Of the three companies, Epic Bio is the only one that has disclosed their development pipeline, which relies on their Gene Expression Modulation System (GEMS) platform. Therapeutics in the research phase at Epic Bio include treatments for: facioscapulohumeral muscular dystrophy, targeting re-methylation the D4Z4 region to suppress DUX4 expression; Retinitis Pigmentosa 4, suppressing endogenous mutated RHO expression and producing normal RHO expression; Retinitis Pigmentosa 11, restoring PRPF3 expression to normal physiological levels; Alpha-1 Antitrypsin Deficiency, suppressing endogenous mutated A1AT expression and producing normal A1AT; and finally, heterozygous familial hypercholesterolemia, targeting pathways known to reduce cholesterol.

## 6. Emerging Applications of CRISPR-Cas in RNA Editing

### 6.1. RNA Editing

The therapeutic potential of genome editing for the treatment of genetic disease is considerable and still emerging. However, unresolved concerns regarding irreversible genetic changes at “off-target” loci may hinder some clinical applications. Recent developments in the identification of RNA-targeting Cas enzymes may offer a path to transient therapies that leverage the precision of CRISPR-Cas technology without the liabilities of permanent/heritable genetic alteration [223].

Enzymatic RNA editing as a biological process was first described in the 1980s–1990s with the discovery and characterization of two main RNA editing protein families, ADAR (adenosine deaminases acting on RNA) and APOBEC (apolipoprotein B mRNA editing catalytic polypeptide-like) [224,225]. The ADAR class of enzymes target specific adenosines in double-stranded RNA (dsRNA) for deamination, thereby converting them to inosines [226]. Inosine base pairs with cytidine and is recognized by translational and splicing machinery as guanosine, resulting in an effective A-to-G substitution [227]. Similarly, the APOBEC family of proteins target cytidines in single-stranded RNA (ssRNA) and ssDNA for deamination to uracil, effectively causing a C-to-T substitution. ADAR editing of coding sequences in pre-mRNAs in the nervous system and dsRNAs that feed into interfering RNA (RNAi) pathways provide important functional diversity beyond what is encoded in the genome [226]. APOBEC1 editing of endogenous mRNAs has been demonstrated to regulate functionally consequential changes to the sequences of encoded proteins, while cytidine deamination within the genomes of RNA and ssDNA viruses by APOBEC3 family members serves as a part of the innate immune response to viral infection [228].

By the mid-1990s, companies and academic groups were already exploring approaches to harness RNA editing for therapeutic benefit. In 1995, Ribozyme Pharmaceuticals demonstrated efficient and targeted reversion of a premature stop codon in dystrophin by endogenous ADAR in *Xenopus* embryos and mammalian cell extracts through the delivery of exogenous RNA oligos [229]. However, due to the imprecise nature of ADAR editing among other considerable technical challenges, RNA base editing as a therapeutic approach has hitherto failed to make significant progress toward advancing into the clinic. In the meantime, the development of zinc finger nucleases, TALENs, and, ultimately, CRISPR has led to an explosion of DNA editing approaches being applied to the development of new therapies and their clinical applications [230]. Other RNA-based approaches such as RNAi and mRNA have also moved rapidly into clinic with multiple blockbuster drugs approved [231]. Nonetheless, the potential to target RNA-mediated diseases and concerns regarding irreversible off-target effects of genomic DNA editing by CRISPR highlights the value that efficient RNA editing technologies could have for clinical and diagnostic use.

### 6.2. Cas RNA Endonucleases

In 2015, the RNA editing field received a jump-start when Cas13 (formerly known as C2c2) from *Leptotrichia shahii* was first described as a programmable RNA-guided RNA endonuclease [232,233]. The Cas13 family of proteins is targeted to ssRNA by sequence-specific 60–66-nucleotide-long CRISPR RNAs (crRNAs), whereupon it cleaves the target and then proceeds to indiscriminately cleave other ssRNA molecules. This activity has already been leveraged for the development of RNA-based COVID-19 and circulating tumor RNA/DNA diagnostics [234]. Base editing technologies relying on fusion of catalytically dead Cas13 (dCas13) to ADAR have also been developed and used to correct disease associated mutations at the RNA level in cellular models [15,235]. However, the in vivo therapeutic potential of Cas13 is limited by considerable collateral cleavage of off-target ssRNAs, which provides a desirable signal amplification for diagnostic purposes, but it is cytotoxic in multiple mammalian cell types [236]. As a result, there is a recognized need for more specific RNA editing tools.

The discovery of Type III-E CRISPR-Cas systems, first characterized in two 2021 publications, appears to be the breakthrough that addresses many of these short-comings [16,237]. Alternately referred to as Cas7-11 by Özcan and colleagues, and gRAMP (giant repeat-associated mystery protein) by van Beljouw et al., these fusions of four Cas7 subunits to a single Cas11 subunit result in the largest CRISPR-Cas effectors yet identified at 1300 to 1900 amino acids. Unlike Cas13, Cas7-11 exhibits highly specific and non-cytotoxic RNA targeting in cells [16]. In initial studies, Cas7-11 supports robust transcript knock-out in mammalian cells and, by fusion of catalytically dead dCas7-11 to ADAR2, efficient RNA base editing [16]. If the prohibitive size of these enzymes can be overcome (some progress in this regard has already been made [238]), this technology opens several new doors for therapeutic applications that could have a significant impact on the rapidly evolving RNA therapy space.

### 6.3. Therapeutic Applications of RNA-Cleaving Cas Enzymes

In addition to many of the same diagnostic approaches to which Cas13 has already been applied, the specificity of Cas7-11 lends itself to in vivo applications for the treatment of cancer, rare genetic diseases, and viral infections (Figure 5). Early studies indicate that Cas7-11 could be effectively used in the same manner as RNAi to negatively regulate gene expression through transcript destruction [16]. Modified Cas7-11 fusion proteins and other approaches that promote transcript stabilization or modulate splice sites can also be envisioned as future variations. Transcript regulation in this manner has broad potential for cancer treatment by modulating the expression of therapeutic targets in a tunable and temporally restricted manner that does not rely on the development of small molecule ligands. This technology could equally be applied in the treatment of rare genetic diseases to control the production of toxic proteins as in Huntington’s Disease, or in modulating any other RNA-mediated process including (but not limited to) hypertension, pain, and cell–cell communication [239,240,241]. The fusion of Cas7-11 to ADAR2 has already been used for efficient base editing in mammalian cells, suggesting additional applications in the treatment of genetic diseases caused by discrete heritable mutations [16]. Importantly, early research indicates that Cas711 may have fewer off-target effects than other RNAi methods which, if it is a general feature of this system across different RNA targets and cell types, could indicate an advantage over current clinical technologies for the modulation of transcript stability [16]. Outside of targeting endogenously encoded RNAs, RNA-cleaving Cas enzymes could have massive potential in combating RNA viruses and retroviruses such as SARS-CoV-2 and HIV.

While significant hurdles including the large size of the Cas7-11 protein, how to package it into a therapy, and how to deliver it effectively to target tissues will need to be addressed in coming years, RNA-cleaving Cas enzymes as therapeutic and research tools are an emerging area at the forefront of CRISPR technology with transformative potential for how diseases are studied and treated.

## 7. Development of Large Animal Models of Human Diseases

### CRISPR and Preclinical Development

The use of CRISPR technology has revolutionized not only the treatment of human diseases but also the field of gene editing in large animal models. Genetically modified large animal models are of significant importance because they are used in the development of new therapies and as research models of human diseases. They are used to demonstrate the feasibility of in vivo CRISPR editing of somatic and germline cells as a therapeutic approach, and to evaluate pre-clinical efficacy and safety of new drugs prior to the initiation of clinical trials [242].

Small animals such as rodents are the predominant models used to study human disease, but often they do not fully recapitulate the pathological changes or symptoms that a disease produces in humans. These differences are due to high anatomical and physiological divergence between species and have led to the failure of many drugs in clinical trials that were only screened in small animal models [243]. Large animal models of human diseases can more accurately recapitulate the characteristics of the diseases, potentially making treatments developed in large animal models more likely to work in humans. Additionally, imaging tools for the screening of the disease or the treatment itself can be tuned in large animal models because of their similarities with humans.

The main large animal models currently used to mimic human diseases are non-human primates, pigs, sheep, goats, and dogs [242] (Figure 6). Non-human primates are the preferred model, not only due to their similarity in physiology and genetics, but also because they display cognition and social behaviors as well. The handicap is that they have long pregnancies and, in general, are expensive to maintain. Pigs are also extensively used as animal models and have the following advantages over non-human primates: early sexual maturity, short reproductive cycle, and high number of offspring per litter [244].

The first monkey genetically modified using CRISPR was reported in 2014 by Niu et al. [245]. Here, two genes were modified simultaneously (peroxisome proliferator-activated receptor gamma, *PPARG*, and recombination-activating gene 1, *RAG1*) by injecting Cas9 mRNA and sgRNAs against the two targets into one-cell-stage embryos. Following this report, Chen et al. [246] showed that CRISPR-genetically modified monkeys could transmit the modification through the germline, and that it was possible to create a model of Duchenne muscular dystrophy by disrupting the dystrophin gene in monkeys [247]. They showed that these monkeys displayed muscle degeneration and other characteristics of the disease in humans. Examples of CRISPR-edited monkeys created to resemble human diseases are the early-onset Parkinson’s disease (PD) model created by Yang et al. [248,249] targeting two exons in the *PINK1* gene, the acute monkey PD models developed by Li et al. [250], Sun et al. [251], and Yang et al. [252], and the model for human adrenal hypoplasia congenita (AHC) and hypogonadotropic hypogonadism (HH) reported by Kang et al. [253] after knocking out the *DAX1* gene. Genetically modified monkey models of human diseases have been also used to evaluate the efficacy of experimental drugs. An example is the study of Tu et al. [254], where they found that the use of the antidepressant fluoxentine treatment alleviated the abnormal brain activities of the autism spectrum disorder monkey model [255].

The first CRISPR-genetically modified pig was reported in 2014 [256] and was created by microinjection of the CRISPR-Cas9 system into the zygotes, making the modification germline transmitted. Another way of creating genetically modified pigs is by somatic cell nuclear transfer (SCNT), where pig fibroblasts can be edited in single or several loci at the same time using CRISPR and then fused into enucleated oocytes to create an edited zygote [257,258,259]. This technology avoids mosaic mutations and detectable off-target effects (see below). Examples of pig models that recapitulate human diseases are phenylketonuria (PKU) [260], Huntington disease (HD) [261], Neurofibromatosis type I [262], and type II collagenopathy [263]. Due to the similarity of organ size between pigs and humans, significant efforts have been invested in the creation of pigs carrying organs that can be transplanted into humans (xenotransplantation). There are two problems associated with this: (1) the immunological compatibility between the two species, or the necessity to immunocompromise the recipient, and (2) the transmission of pig viruses (porcine endogenous retroviruses or PERVs) into humans. To overcome these problems, several KO pig models lacking the rejection antigens [264,265,266] or carrying inactivated PERVs have been developed [267,268], all mediated by CRISPR. These breakthroughs have led to the first transplants of pig heart and kidney into humans [269,270].

The CRISPR-gene-edited sheep and goats that have been produced are related to the agriculture and pharmaceutical fields. Those two species are of high value because of their meat, milk, fibers, and other bio-products, so the gene modification efforts (zygote injection of CRISPR-Cas9, or SCNT with CRISPR-modified fibroblasts) have been applied preferentially into those. Despite that, there are several sheep models for human diseases such as cystic fibrosis, created by disrupting the *CFTR* gene [271], human hypophosphatasia, developed by knocking out the *ALPL* gene [272], and human deafness, achieved by editing the *OTOF* gene [273].

There are few groups working to create genetically modified dogs that resemble human diseases using the CRISPR/Cas9 system. One prominent dog study used CRISPR-Cas9 (delivered by adeno-associated viruses AAV) to restore the expression of dystrophin in a dog model of Duchenne muscular dystrophy, accompanied by improved muscle histology and amelioration of the disease [274]. Several models have been created by knocking genes into the genome of large animals. For example, a non-human primate model with OCT4-hrGFP (octamer-binding transcription factor 4-humanized recombinant green fluorescent protein) was achieved by using CRISPR-Cas9-assisted HR [275]. The pig model of HD mentioned above was developed by Yan et al. [261] using CRIPSR-Cas9 to insert a large CAG repeat into the pig *HTT* gene in fibroblast followed by SCNT. The brain of this pig model showed neurodegeneration in the medium spiny neurons like affected patients do. One example of sheep created by knocking in using CRISPR is the turbo GFP sheep [276].

There are several advantages of using CRISPR-Cas9 to create large animal models. The first is that this system is specific and can target any gene in the genome. The second is that several genes can be edited at the same time by co-injecting/delivering different gRNAs in one-step, which makes the development of human multigenic disease models possible. The third advantage is that the creation of the homozygous mutant founders of a colony is much faster because CRISPR can edit both alleles, so investigators do not have to mate heterozygous founders to obtain the KO animals. This advantage is critical because it saves years of mating large animals to establish your model of interest. However, some technological challenges remain. One is the production of off-target effects. If the gRNA sequence has high similarity with another part of the genome than the intended target, off-target editing can cause undesirable effects in the model. If off-target effects happen, they could be diluted over generations, but it can take years in the case of large animals due to their long gestational period and time to reach sexual maturity. The risk of off-target effects can be reduced using publicly available bioinformatic tools to optimize gRNA design. Another challenge is the possibility of mosaicism or having an animal model with a mix of cells with different genotypes. This issue could happen for several reasons: (a) injection of Cas9/gRNA is too late, when the zygote has already started dividing, (b) the translation of the Cas9 mRNA is delayed and does not happen until the embryo has divided already, or (c) the expression of Cas9 mRNA is prolonged. Although mosaic animals can help us understand the dosage effect in some diseases, it is undesired in most studies. To overcome it, there are several strategies than can be implemented: (a) inject the CRISPR components in a more efficient format, like Cas9/gRNA ribonucleoproteins, so they are ready to act because they do not need to be translated, (b) use electroporation to deliver the components into early one-cell zygotes before the first division happens [277], and (c) shorten the half-life of the Cas9 enzyme [278].

In terms of the market, there are five main companies producing large animal models: Recombinetics Inc. (Eagan, MN, USA) (with its four subsidiaries, Surrogen, Makana Therapeutics, Acceligen, and Regenevida), Precision BioSciences (Durham, NC, USA) (partnering with Agrivida, Inc., Woburn, MA, USA), eGenesis, Genus plc (Cambridge, MA, USA) (together with Caribou Biosciences, San Diego, CA, USA), and Synthetic Genomics/Lung Biotechnology. They also focus on editing the pig genome in the way they can produce human transplantable organs.

More recently, CRISPR-Cas systems have been modified to generate missense mutations and early stop codons. One example is the third-generation base editor (BE3), which has been used by several investigators to create large animal models that resemble different human diseases. Li et al. (2018) used BE3 and SCNT to develop one pig model for the ablepharon macrostamia syndrome (AMS) in humans, which is caused by a point mutation in the *TWIST2* gene, and another pig model for the oculocutaneous albinism type 1 (OCA1), caused by mutation of the *TYR* gene [279]. Both models reproduced very accurately the phenotypes of human diseases. Another example of the use of BE to create an animal model was reported by Wang et al. [280], where they showed the first Hutchinson-Gilford progeria syndrome (HGPS) monkey model that was achieved by mutating the *LMNA* gene via microinjection into zygotes. Other groups have used BEs to prove that different point mutations in several genes can be achieved in one animal model. A group of researchers used cytosine base editors (CBEs) delivered by embryo injection or SCNT to create pigs carrying point mutations in three different genes at the same time (*DMD*, *TYR*, and *LMNA*; *RAG1*, *RAG3*, and *IL2RG*) [281]. In this study, they also achieved the edition of multiple copies of the PERV in porcine embryos and fibroblasts. Yuan et al. [282] also utilized BE3 and successfully produced pigs that had three-point mutations in the genes *GGTA1*, *B4galNT2*, and *CMAH*.

## 8. Conclusions

Ever since its inception, the CRISPR-based genome editing technology has rapidly transformed human medicine. In the field of cancer, FDA approvals of multiple adoptive cell therapies in the last decade have heralded a paradigm shift in the way cancer is treated. The co-emergence of CRISPR technologies and its coupling to cell therapy has enabled rapid improvements in the cell therapy space. Using CRISPR-Cas9, precision-tailored and multiplexed gene editing is enabling the generation of durable and efficacious off-the-shelf allogenic therapies for several cancers without side effects from fratricides, CRS, etc.

Many of the CRISPR-Cas9-edited cell therapy products are still in early-stage clinical trials, and given the small number and limited duration of the current clinical trials where therapeutic products with low gene editing efficiencies were tested, the need of the hour is for longer-duration trials testing therapeutic products with higher gene editing efficiencies to better understand the safety of the CRISPR-edited cell therapy products. Forthcoming results from several in-progress clinical trials will provide critical insights for development of future CRISPR therapeutics. Beyond cancers, the power of the CRISPR-Cas system has facilitated the development of precision tools for the in vivo demolition of genome-integrated viruses such as HIV, HPV, and HSV-1 in tissues. Likewise, it shows promise for those suffering from life-long, debilitating, and/or degenerative diseases such as sickle cell disease and β-thalassemia, type 1 diabetes, and Duchenne muscular dystrophy, opening the door for the previously impossible: not just a treatment for, but a functional cure for these diseases.

The application of CRISPR-Cas technology to generate large animal models which mimic human diseases is of significant importance, as they facilitate the safety and efficacy of testing prior to human application. Several different diseases (cystic fibrosis, Duchenne muscular dystrophy, and Parkinson and Huntington diseases, to name a few) have been modeled using monkeys, pigs, sheep, goats, and dogs, and they have the advantage of having short reproductive cycles and similar physiology and organ size to humans, thus enabling accelerated testing during therapeutic product development.

Novel approaches for therapeutic development that rely on CRISPR-epigenome editing are still in the research phase and offer the promise of inducing heritable changes in gene expression without relying on DNA cutting and error-prone repair. In addition, other in vivo therapies are being developed that directly modulate or bypass disease-causing alleles, sometimes in a tissue-specific manner, which show great promise but suffer from the similar concerns with off-target effects. Emerging technologies using RNA-targeting Cas enzymes may be an additional approach to leverage the specificity and versatility of CRISPR-Cas in a reversible manner that obviates the need to deliver gene-edited material. Further development of this technology will likely be necessary before it is clinically impactful.

Taken together, the CRISPR-Cas system has transformed within a decade into a product and technology platform with many different tools in its arsenal. As detailed in this paper, each of these tools has a unique role to play, all the way from lab bench to bedside, and one can only predict that the CRISPR-Cas system will evolve to be better and become more potent in its second decade of invention.

## Figures and Tables

**Figure 1 cells-12-01103-f001:**
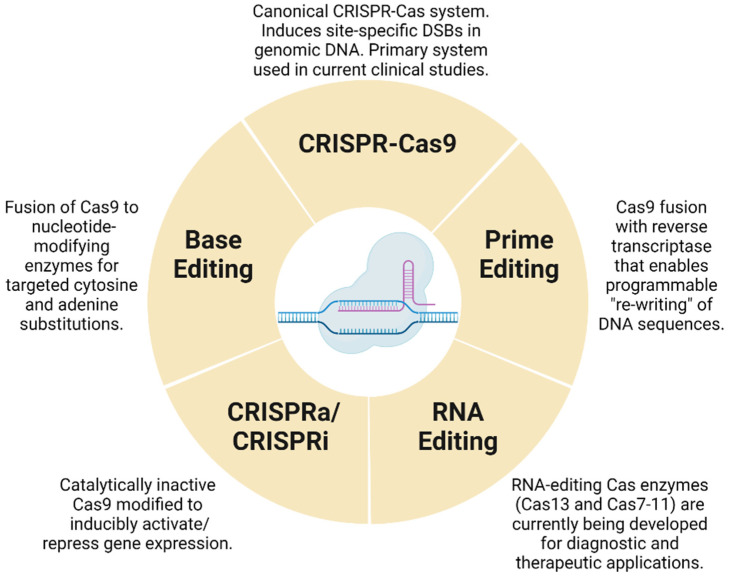
Summary of the various CRISPR-Cas systems discussed in this review.

**Figure 2 cells-12-01103-f002:**
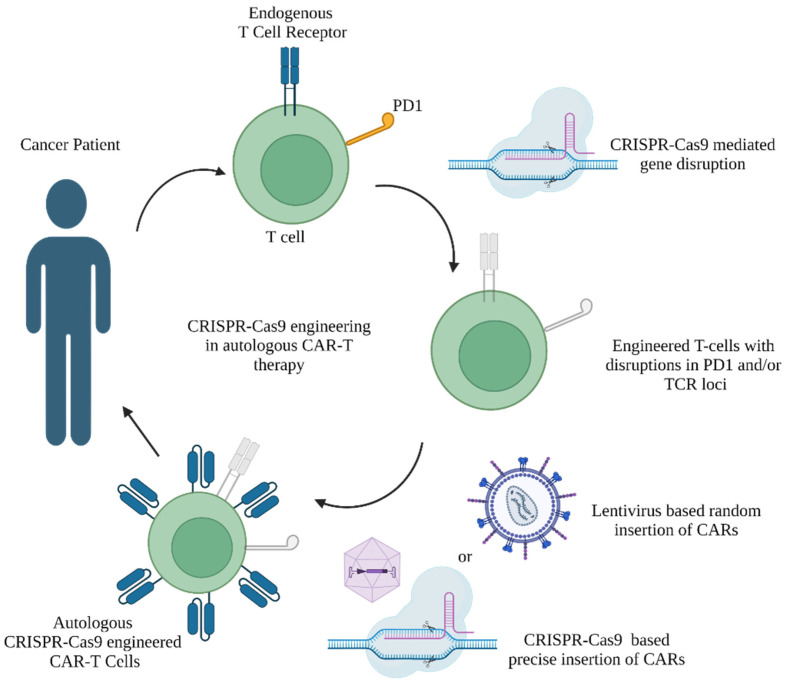
A scheme of CRISPR-Cas9 engineering in autologous CAR-T therapy. T cells are extracted from cancer patients and engineered with CRISPR-Cas9 to edit select genes. Later, chimeric antigen receptor (CAR) is randomly inserted into the genome via lentivirus or precisely engineered into a target locus via the CRISPR-Cas9 system to create an engineered autologous CAR-T cell. Engineered CAR-T cells are later given back to the cancer patients.

**Figure 3 cells-12-01103-f003:**
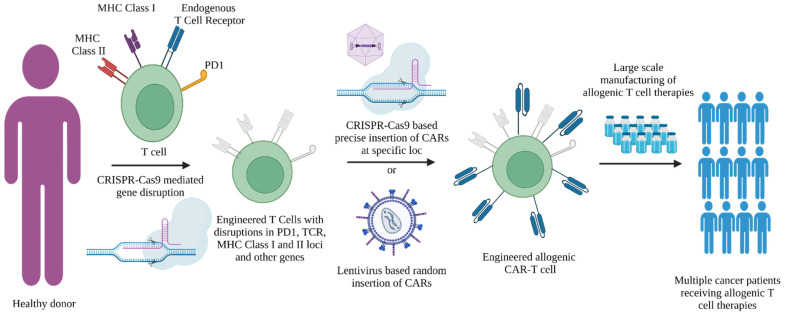
A scheme of CRISPR-Cas9 engineering in the allogenic CAR-T therapy. T cells are extracted from healthy donors and engineered with CRISPR-Cas9 technology to edit select genes including genes for TCR, PD1, and MHC Class I and II molecules, among others. Later, CAR is randomly inserted into the genome via lentivirus or precisely inserted at a target locus via the CRISPR-Cas9 system to create an engineered allogenic off-the-shelf CAR-T cell. These donor-derived allogenic CAR-T cells are manufactured in scale and given back to multiple cancer patients.

**Figure 4 cells-12-01103-f004:**
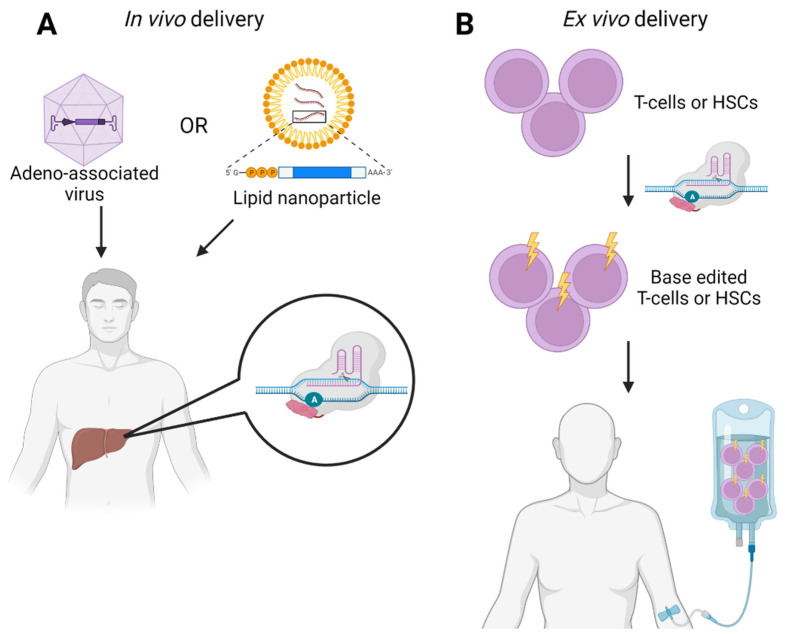
Delivery systems for base editors. (**A**) Base editors can be delivered directly to patients in vivo, typically through adeno-associated viruses or lipid nanoparticles. The base editor then acts directly in patients (e.g., VERVE-101 edits *PCSK9* in the liver). (**B**) Base editors can also be used to modify T cells or HSCs ex vivo before the edited cells are infused into patients.

**Figure 5 cells-12-01103-f005:**
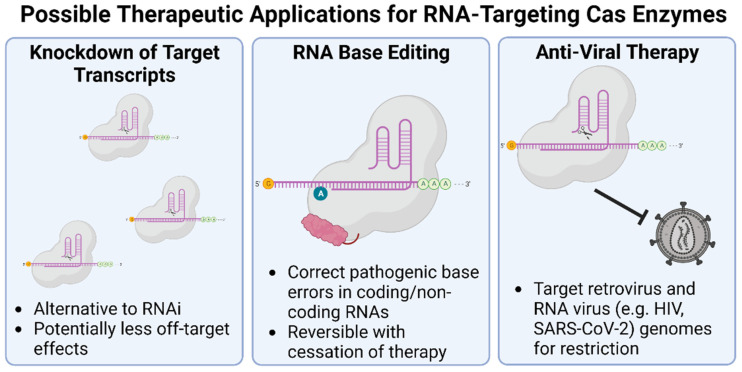
Emerging therapeutic applications for RNA editing by Cas enzymes.

**Figure 6 cells-12-01103-f006:**
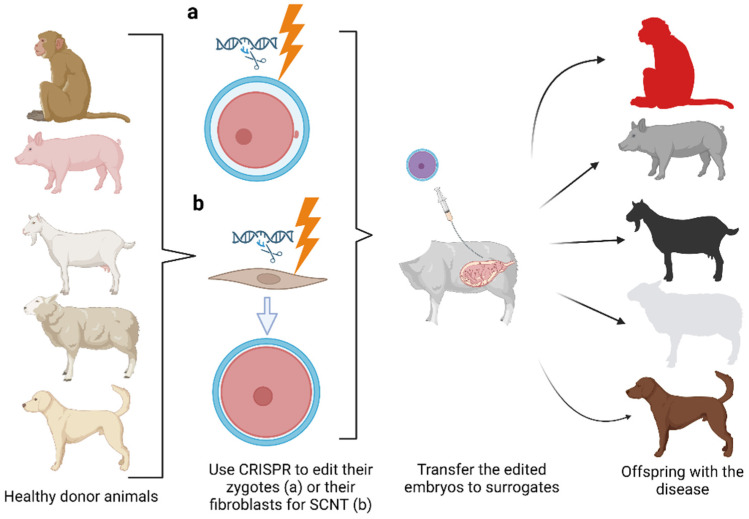
Diagram showing the steps to generate large animal models. SCNT: somatic cell nuclear transfer.

## Data Availability

No new data were created or analyzed in this study. Data sharing is not applicable to this article.

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
