# Peer review of "CRISPR-Cas System: The Current and Emerging Translational Landscape"

_cells, 2023, doi:10.3390/cells12081103_

Round 1

Reviewer 1 Report

This review comprehensively discussed the development of CRISPR-Cas system, its application in clinical treatment, and the state-of-the-art CRISPR-Cas9 technology. However, the article is not well-organized and has inconsistent text, which needs to be polished.

Here are some general suggestions:

1. Most of the text review the development of the CRISPR/Cas system in the cell or animal models (section 3 to 7) but not the therapeutic product. I suggest the title needs to be changed.

2. The way to deal with the abbreviation is inconsistent, for example, in some terms that use capital for each word before the abbreviation (for example, line 177 and line 191). Some are not (for example, line 105 and line 189). It must be consistent in the whole text (for example, you mention TILs is tumor Infiltrating Lymphocytes in both line 105 and line 177).

3. Some of the text has capitalization (for example, line 286 and line 292). Is it for any reason?

4. Some titles have a colon, and some do not. Please check and be consistent.

5. Remember to show the full name before the abbreviation (for example, line 324: KO)

Below are other suggestions according to the specific line in the text:

Line 16 The full name of CRISPRi /CRISPRa should be included.

Line 25 In the text, sometimes the system is called “CRISPR-Cas system” and sometimes are “CRISPR-Cas9 system”. Please be consistent.

Line 27 “CRISPR-Cas9 is comprised of two elements: an RNA-guided DNA endonuclease called Cas9 isolated from Streptococcus pyogenes and its guiding RNA called single guide RNA or sgRNA” According to the reference, CRISPR-Cas9 system has 3 elements: sgRNA, tracrRNA, and Cas9.

Line 65 “Within its first decade…” probably should cite the original paper in 2013, same as line 97.

Line 130 “his” should be “this,”

Line 138 Should it be PD1 or PD-1? Or is there any difference?

Line 140 the abbreviation should be PDL1.

Line 148 has multiple dots.

Figure 1. CRSPR-Cas9 may target either PD-1 or TCR but not always both at the same time, so the tag in the illustration should be “or/ and” instead of “and”.

Line 153 “… results from three Phase 1 trials are published…” but later in line 155 you mention the fourth trial (Ref. 53). So total may have 4 trials that passed clinical phase 1 were published? Or the citations in line 153 were wrong?

Line 157 Since the text mentions the longer T cells persisting time, how long did the engineered T cells survive in previous trials?

Line 163 Is prolonged persistence of engineered T cells positively correlated to anti-tumor activity?

Line 167 Again, how long did the engineered T cells survive in the previous trial? Are 4 weeks shorter than the previous trials?

Line 168 “PD1 disrupted cells persisted in patients for 4 four weeks after infusion…” The “4 four” need to be corrected.

Line 177 You already identified tumor infiltrating lymphocytes as TILs in line 105, it is not necessary to mention it again.

Line 311 “…indicating the low editing frequency of CRISPR-Cas9 system in this study.” What’s the average or general editing efficiency in T-cells? Compared to ref. 51 (5.8% editing efficiency) the efficiency in this study is still high.

Line 312 “There were off-target effects for each locus but their frequencies were very low.” The sentence is vague. How low are the off-target frequencies, and what are the general off-target frequencies for these loci? Same as in line 314.

Line 359 What is the point to named it “exa-cel” and CTX001?

Line 361 What is “hHSPCs” stand for?

Line 363 The citation should be “Khosravi et al., 2019”. The way to cite the illustration may not be a good way.

Line 380 The text “…through Cas12a RNP-mediated disruption…” conflicts with the text in the same paragraph on line 375 “…using CRISPR-Cas9 gene editing therapies…”

Line 452 Should have the reference from Dr. David Liu’s group.

Line 489 The NCT05398029 did not have a reference citation.

Line 515 The NCT05397184 did not have a reference citation.

Line 521 Ref. 123 is news that may not be suitable for a scientific article.

Line 558 You should have the full name of CRISPR interference and activation in the title.

Line 669 Table 4 does not have NCT249.

Line 675 Is CRD stands for Cure rare disease, Inc? 

Line 686 Why switch to the FDA.gov database but not the clinicaltrial.gov where this review generally did?

Lin 730 “…Chroma Medicine, co-founded by Jonathan Weissman, David Lui, and Keith Joung…” The founder should be David Liu.

Line 853 “The main large animal models currently used to mimic human diseases are non-hu- 853 man primates, pigs, sheep, goats, and dogs.” Any reference for this statement?

Line 872 and line 874 have two extra left brackets.

Line 896 Reference 240 only published the xenotransplantation of pig kidneys.

Line 905 “There are not many groups working to create genetically modified dogs that resemble human diseases.” The statement conflict with line 853: “The main large animal models currently used to mimic human diseases are non-human primates, pigs, sheep, goats, and dogs (Figure 5).”

Reviewer 2 Report

The authors reviewed the therapeutic products developed using CRISPR-Cas, including therapies currently in clinical trials and large animal models of diseases. Furthermore, they include new CRISPR-based tools that are promising therapeutic products.

There are many already published works reviewing CRISPR-Cas related clinical trials. However, the do not include new tools or animal model, making this review more complete, interesting, and worth for publication.  

The review is correctly organized, but the quality of the sections is not homogeneous, being some of them well explained and discussed while others lack depth. Because of that and other details explained bellow, the manuscript must be improved to be accepted for publication.

-Major concerns:

1. Sections 2.2, 2.3 and 5: They are mainly a list of current clinical trials, lacking explanation, analysis and discussion. These sections are a big contrast in comparison with the other sections, which are well written and have the depth required in a review. Therefore, these sections should be rewritten.

2. Section 5.3.6: Section 5 is focused in the use of CRISPRi/CRISPRa and epigenetic editors. However, the only clinical trial presented, in section 5.3.6, is not related with this kind of tools. It used active Cas9 RNP, targeting 2 sites flanking the mutated exon, for exon skipping. Here CRISPR is not used to induce or repress gene transcription; it is just deleting a stop codon that was impairing the expression of full-length dmd protein. Thus, this clinical trial must be included in section 2.2. Furthermore, it should be better explained.  

3. Section 2.2 (line 346): “other non-infectious diseases” is not correct.

4. There is a lack of figures explaining the different CRISPR tools: Introduction: Figure of CRISPR systems reviewed here. Section 4: Figure explaining prime editing.

-Minor concerns:

5. Table 1, 2 and 3 must be in the same format.

6. Section 2.1. The manuscript shows that all clinical trials using CRISPR for cancer therapy are based on immunotherapy. It would be nice to have a small overview of pre-clinical studies using CRISPR as a direct therapy on cancer cells, and the challenges and perspectives of these strategies.

Round 2

Reviewer 2 Report

The authors addressed all the previous concerns, and the manuscripr is now ready for publication.

Author Response

We thank the reviewer for the comments.